# Liability and Medico-Legal Implications in Estimating the Likelihood of Having Attained 14 Years of Age in Pediatric Clinical Practice: A Pilot Study

**DOI:** 10.3390/healthcare11233047

**Published:** 2023-11-26

**Authors:** Roberto Scendoni, Dora Mirtella, Roberto Cameriere, Luca Tomassini, Francesco De Micco, Francesca Baralla, Mariano Cingolani

**Affiliations:** 1Department of Law, Institute of Legal Medicine, University of Macerata, 62100 Macerata, Italy; r.scendoni@unimc.it (R.S.); dora.mirtella@unimc.it (D.M.); mariano.cingolani@unimc.it (M.C.); 2Department of Medicine and Health Sciences, University of Molise, 86100 Campobasso, Italy; roberto.cameriere@unimol.it (R.C.); francesca.baralla@unimol.it (F.B.); 3International School of Advanced Studies, University of Camerino, 62032 Camerino, Italy; 4Research Unit of Bioethics and Humanities, Department of Medicine and Surgery, Università Campus Bio-Medico di Roma, 00128 Roma, Italy; f.demicco@unicampus.it; 5Department of Clinical Affairs, Fondazione Policlinico Universitario Campus Bio-Medico, 00128 Roma, Italy

**Keywords:** age estimation, liability, pediatricians, orthopanoramic dental radiographs, clinical practice and treatment

## Abstract

Accurate methods of age estimation are more essential than ever due to the rise in undocumented individuals without proper identification, often linked to illegal immigration and criminal activities. This absence of reliable records presents challenges within the legal systems, where age thresholds in the context of children’s rights vary across countries. Age 14 has global significance, as established by the UN Convention on the Rights of the Child and the EU for administrative purposes. Accurate age estimation is crucial in medical decisions, reproductive health, and forensics. This study focuses on age estimation via dental radiographs, proposing a method for estimating the likelihood of having attained the age of 14. Orthopantomograms were analyzed from two samples, 191 Italian children (aged 5–15) and 822 Chilean subjects (aged 11–22), using dental maturity indices. These indices evaluated open tooth apices and complete root development. Statistical analysis confirmed the method’s reliability in identifying individuals aged 14 or older, with sex-specific cut-offs. The proposed method particularly advocates an approach based on dental mineralization, which could surpass those relying on bone growth. The collaboration between medical experts, including pediatricians and diagnostic imaging specialists, is vital for standardized age estimation strategies. Ethical concerns regarding radiation exposure and accountability are recognized, although the method’s low radiation doses are deemed acceptable. The proposed method will help health professionals to accurately predict whether or not the 14-year threshold has been reached, opening up new avenues of medico-legal interest and laying the foundations for a legal framework that would allow the pediatrician, when involved, to use a valid and recognized diagnostic approach.

## 1. Introduction

In today’s society, the need to determine the age of living persons is becoming more crucial every day because of the progressively high number of people without ID cards or regular birth records who immigrate illegally or commit crimes. In the case of people without regular ID documents, it is difficult to determine the exact age of the person when they come into contact with the judicial system. In particular, Europe has seen a significant increase in the number of irregular migrants in recent years, and children have made up a large proportion of those who have made the journey, either with families or alone [1].

In this context, the age limits for assigning a person certain rights (e.g., to vote, to work, to drink) vary between countries. Furthermore, 14 is the most commonly accepted age of complete criminal liability across the world, according to the UN Convention on the Rights of the Child, which was adopted in 1989. More recently, the European Union (Council of Europe Resolution of 26 June 1997) also regulated the age assessment of unaccompanied minors for administrative and social purposes [2].

Age estimation may concern pediatricians when determining the proper administration of age-appropriate drugs, i.e., whether to prescribe pediatric drugs or drugs for adults or deciding that a drug is contraindicated. A further topic of interest concerns whether the child or adolescent’s opinion should be taken into account in medical decisions; it seems widely agreed in most countries that they should be involved from age 14. In Italy, according to the law, the age of consent to sexual activity is 14. Contraception (including emergency contraception) is legally accessible to minors over 14 even without the consent of their parents or guardians. Moreover, according to the law, minors may give autonomous consent to abortion if they are recognized as a mature adolescent and at least 14 years old. 

In the forensic field, physical developmental charts and social parameters are used to estimate the age of a person. Furthermore, forensic age estimation, which often employs radiological methods, presents ethical concerns and emphasizes the need for standardized strategies. This multidisciplinary endeavor, involving specialists in pediatric medicine, diagnostic imaging, and others, aligns with evolving regulations to ensure accuracy and fairness, with regard to both healthcare professionals and children undergoing investigations.

In recent years, several methods for age estimation using orthopanoramic dental radiographs have been tested in different populations with positive results [3,4,5]. 

In the present study, we aim to provide pediatricians with an “age estimation” method for predicting the attainment of the age threshold of 14 years: we believe that the topics mentioned may be useful in pediatric practice.

## 2. Materials and Methods

This study is based on the analysis of orthopantomograms (OPGs) that had been performed for clinical reasons, and only the sex and age of the subjects were known. In order to provide a method to determine the attainment of age 14, we analyzed OPGs from two samples: a group of 191 native Italian children between 5 and 15 years old (102 boys, 89 girls), and another sample of 822 Chilean nationals aged between 11 and 22 (472 girls and 350 boys). 

For all subjects included in this study, predefined exclusion criteria comprised tooth irregularities, hypodontia, systemic illnesses, and orthodontic interventions that could influence dental development, including third molars. Pertinent details, such as patient ID number, sex, birthdate, and X-ray date, were meticulously documented. The chronological age of the patients was recorded in decimal years.

All participants had been undergoing orthodontic treatment and exhibited normal growth patterns without any indications of growth disorders. The radiographs had been captured as a standard procedure over a five-year period.

The chronological age of each individual was determined by calculating the time elapsed between their date of birth and the date when the X-rays were taken. These data points were meticulously recorded in a Microsoft Excel^®^ spreadsheet.

Following ethical and deontological principles, informed consent was obtained for the collection of OPGs. We used de-identified data, reducing the risk of any individual patient’s identity being linked with the data. In the case of minors, informed consent was obtained from parents and/or legal tutors, while for adult subjects, consent was obtained directly from the individuals themselves.

The purposes of the study were explained in the consent form, and it was clarified that data collection would be intended for scientific research purposes, including articles, presentations, or other academic publications, guaranteeing data security at all times.

The sex and accurate age of all participants were documented at the inception of the study. The OPG images exhibited easily discernible and quantifiable third molars. 

We performed randomization tests on both samples, extrapolating 2/3 of the Italian cohort to a broader range of subjects within the age bracket of 5 to 15 years. The same procedure was applied to the Chilean cohort within the age range of 11 to 22 years, also derived from a larger dataset.

Clinical data were used to detect the best cut-off for 14 years of age. The data used for the survey were processed anonymously and collected by healthcare professionals with prior informed consent, in compliance with laws on confidentiality and protection of personal data. If at least two permanent lower molars were not completely developed, we selected a cut-off following the study on Italian children [6]. The ratio between the distance between the inner sides of the open apex and the tooth length of seven permanent teeth of the left mandibular were measured (Figure 1) [6,7]. 

Age estimation was evaluated by the sum of normalized open apices and numbers (N0) of teeth with complete root development [7]. 

Specifically, seven left permanent mandibular teeth were assessed.

### 2.1. Intra-Observer and Inter-Observer Agreement

To establish a uniform approach, 30 X-ray images were collectively reviewed by the observers to guarantee the consistency and replicability of this research. The first author meticulously analyzed each digital panoramic radiograph on three separate occasions (at four-week intervals) using the CS Imaging software version 8 to verify internal consistency. Additionally, a second evaluator employed the same technique to scrutinize 10% of the overall datasets (n = 101) to authenticate external reliability. The two datasets were compared by means of the intraclass correlation coefficient scores for intra- and inter-observer measurements.

### 2.2. Statistical Analysis

Once a cut-off, c, was established for a given maturity index, S, we considered the individual to be 14 years of age or older if S was lower than c, i.e., the test was positive. In this case, the sensitivity p1 of the test (i.e., the proportion of subjects older than or equal to 14 years of age with S < c) was evaluated together with its specificity p2 (i.e., the proportion of individuals younger than 14 with S ≥ c). The dental maturity index may help to discriminate between individuals who are above and below the age 14 threshold by the post-test probability of being 14 years of age or more (i.e., the proportion of individuals with S < c older than or equal to 14 years). According to Bayes’ theorem, the post-test probability may be written as: $$p=\frac{p_{1}p_{0}}{p_{1}p_{0}+(1-p_{2})(1-p_{0})}$$
where *p* is a post-test probability, and *p*_0_ is the probability that the individual in question is 14 or older. Statistical analysis of data and related graphs was carried out by the R statistical program version 3.6.1 (R Core Team, 2018) [8]. 

## 3. Results

Data analysis produced the following results. If there were at least two open apices excluding the third molar, the cut-off for males was 0.101 with a sensitivity of 80% (CI: 65–91%), a specificity of 89% (CI: 82–94%), and a positive predictive value of 88% (CI: 81–93%; Table 1). If there were at least two open apices excluding the third molar, the cut-off for females was 0.113 with a sensitivity of 83% (CI: 61–95%), a specificity of 92% (CI: 86–96%), and a positive predictive value of 91% (CI: 85–95%; Table 1). Table 1 shows the reference cut-offs for both male and female subjects with at least two teeth with open apices (third molar excluded).

If there were second and third molars with open apices, the cut-offs for males were 0.73 (3M) and 0.16 (2M) with a sensitivity of 84% (CI: 79–89%), a specificity of 90% (CI: 82–96%), and a positive predictive value of 96% (CI: 93–98%; Table 2). If there were second and third molars with open apices, the cut-offs for females were 0.77 (3M) and 0.1 (2M) with a sensitivity of 81% (CI: 76–85%), a specificity of 80% (CI: 72–86%), and a positive predictive value of 93% (CI: 90–95%; Table 2). The reference cut-offs for both male and female subjects with second and third molars with open apices are indicated in Table 2.

### Intra-Observer and Inter-Observer Agreement

The research findings revealed an ICC value of 0.98 for intra-observer reliability and 0.96 for inter-observer reliability. This indicates an outstanding level of agreement between the assessments.

## 4. Discussion

The global health challenge presented by migration is particularly impactful on children. Currently, over 13 million children reside as refugees or asylum seekers beyond their country of origin. A substantial portion of those undertaking these journeys consists of children, whether accompanied by their families or unaccompanied, seeking safety, stability, and a brighter future. From 2015 to 2017, Europe saw the submission of over one million asylum applications on behalf of children, with a significant number originating from Syria, Iraq, and Afghanistan. In 2017, an astonishing 70% of the 210,000 asylum claims for children in Europe were concentrated in Germany, France, Greece, and Italy [1,9].

The phenomenon of migration to Europe is characterized by continuous evolution, with frequent changes in common migration routes, modes of travel, and lengths of stay in transit countries. Children undertaking these perilous and often protracted journeys are exposed to significant health risks. The health of children on the move is intricately linked to their pre-journey health status and the conditions experienced during transit and after arrival and is influenced by their exposure to trauma, the health of their caregivers, and their access to healthcare [10,11].

The majority of the literature addressing the health of migrant children has originated from North America and Oceania. However, given the substantial increase in the number of children arriving in Europe and the necessity for a more comprehensive understanding of their situation in the European context, this article reviews the health risks and needs of children on the move in Europe and assesses how European healthcare policies respond to these challenges and needs [12,13].

The Convention on the Rights of the Child (CRC) grants all children the right to healthcare without discrimination. Specific attention is devoted to the rights of displaced and unaccompanied children in Articles 2, 9, 20, 22, 30, and 39 of the CRC, providing a valuable framework for addressing the health of children on the move [14].

The evolving nature of the migration phenomenon, coupled with the unique vulnerabilities and needs of migrant children, underscores the urgency of tailored healthcare policies and interventions to address the multifaceted challenges they encounter. It is critical to recognize that their health risks and needs are distinct from the local population and vary between migrant groups, necessitating a comprehensive and compassionate response to safeguard their well-being [1,10,13].

The need to study precise and accurate methods of age determination is more essential than ever. The present work aims to illustrate a method to ascertain whether or not a subject is 14 years of age. 

Previous studies have mainly aimed at evaluating the ratio between chronological age and two anatomical areas, such as the bones of the hand/wrist [15,16,17,18,19,20,21,22,23]. These structures were chosen for many reasons: compared with other parts of the body, images can be obtained with lower amounts of radiation, and many bones or teeth are generally available. In the last few years, within the ambit of the AgEstimation Project, several works have been published on the problem of age estimation for forensic purposes in young subjects, including methods using both teeth and the bones of the hand/wrist [24,25,26,27,28]. Pediatricians and auxologists in clinical practice generally use growth curves based on known chronological age to investigate and monitor a patient’s physical development. A different approach must be adopted for the age estimation of pediatric subjects without ID cards from an unknown background. No psychological test can determine an individual’s age; such an assessment can only help to define the degree of maturity. But the law considers that a person reaches maturity at 18 years old, except in special cases. Therefore, the maturity assessment cannot be a useful index. 

Usually, for the age estimations of subjects with unknown chronological ages, pediatricians use methods based on examining skeletal development, in particular the wrist. These methods present significant limits because it has been scientifically demonstrated that a minor’s bone development is influenced by their socio-economic situation and nutritional status [29]. Thus, methods based on bone growth can only yield accurate results in cases where the chronological and socio-cultural history of the minor is known. 

“Age estimation” methods based on dental mineralization have been found to be more reliable because this process is not substantially influenced by diet, health, and clime. Therefore, skeletal age and dental age could be different: in a well-nourished minor, skeletal age and dental age are almost consistent with chronological age, whereas, in an undernourished minor, skeletal age is younger than chronological age [30].

We believe that pediatricians are aware of the potential fallibility of age estimation based on skeletal development and of the greater reliability of age assessments based on dental mineralization.

Therefore, we propose a simple method for estimating the attainment of age 14, which could offer a new approach to age assessment of minors without identification.

As discussed in the Section 3, for both males and females, age estimation based on the presence of open apices in the second and third molars yielded significant results. For males, the method demonstrated high sensitivity (84%) and specificity (90%). Similarly, for females, the method maintained high accuracy, albeit with slight differences compared to males, with a sensitivity of 81% and specificity of 80%. These results indicate that the method is reliable in age estimations based on the specific criteria considered in molars, with a significant ability to correctly identify males and females of 14 years.

In relation to ethnicity, the two groups were considered jointly; however, an assessment of each individual group was performed and demonstrated high sensitivity and specificity with no significant difference compared to the values obtained from the two groups together. This study serves as a trial to gauge the feasibility of assessing the 14-year-old age group using the proposed method. Once this method is established, the authors can subsequently compare it with other methods, such as the Demirjian Method, the ratio of pulp-to-tooth area or even the third molar maturity index. 

The Demirjian method involves separate assessments of the developmental stage for each of the seven permanent teeth (excluding the third molar), and a score is allocated to each stage, from A to H, based on their formation and from the appearance of the first calcification centers to the closure of root apices. The sum of the seven scores generates a “maturity score” convertible into age through gender-specific tables. Although the technique is considered reliable, studies have highlighted its dependence on the characteristics of each population. Nevertheless, the method is widely used in forensic contexts thanks to its practicality and ability to define distinct stages. Demirjian himself revised the method, developing three variants based on radiographs of Franco-Canadian children aged 3 to 17 years: a revised method for seven teeth, one based on four teeth (molars and premolars), and another on the four incisors [31,32,33].

Another method worth mentioning is the Gambier approach. In this case, OPGs are used to analyze the eruption of the third molar (M3). In the context of a study conducted at a French university hospital, 557 OPGs of individuals between 15 and 24 years old were examined. The complete eruption of the third molar in stage 3 (complete emergence in the occlusal plane) was common in 85% of females and 98% of males. Nevertheless, the eruption stages of the third molars did not allow for a reliable distinction between minors and adults for both sexes [34,35].

Therefore, the presented method can be seen as an alternative approach that, if developed, could take center stage in further comparative studies with other methods, such as those described above.

The purpose of this study is to enhance the previously proposed methods by introducing an age estimation technique based on dental mineralization. This approach is less influenced by external factors such as skeletal growth. Additionally, the new method is characterized by precision, practicality, and ease of use.

Furthermore, the application concerns subjects approximately between the ages of 4 and 14 and in any case minors, while other dental methods proposed by Cameriere are used to establish cut-offs regarding the age of majority or to estimate the age of subjects who are already adults [6,7,30]. The open apices method has already been proposed in other works [30]; however, in this case, it was tested on a large number of subjects, which allowed us to refine sensitivity and specificity. Furthermore, our study was applied simultaneously to subjects of different nationalities, a very useful aspect to consider in age estimation research since the geographical origin of a minor may not be known at the time of a clinical assessment.

The presented method requires exposure to radiation, and it is, therefore, crucial that children and young people (or those responsible for them) be appropriately informed. 

From a medico-legal standpoint, the requirement for instrumental assessments to estimate a minor’s age necessitates close collaboration among pediatricians, potentially neuropsychiatrists, and specialists in diagnostic imaging. This collaboration makes age estimation possible through anthropometric indices as a multidisciplinary endeavor. This aspect carries special significance since the team as a whole carries professional accountability, as well as each individual within the team. Challenges regarding the accountability of individuals within a team persist in certain countries, such as Italy, where this matter remains a topic of debate within legal jurisprudence [36]. Furthermore, radiological examinations for age estimation raise ethical concerns that involve the pediatrician primarily, as well as other healthcare professionals, since the minor is subjected to radiation exposure for legal rather than therapeutic purposes. Several age estimation studies have been performed on minors using magnetic resonance imaging (MRI) to circumvent this problem; however, it is more expensive and less usable than X-ray examination [37].

It is important to note that healthcare practitioners engaged in these activities, serving within medical facilities, find themselves intervening as public officials. This underscores the need to adopt standardized strategies for age estimation, given the particular significance this holds within the realm of justice.

Additionally, methods for age estimation in minors, such as the one proposed here, necessitate careful training of the involved personnel to implement these techniques. This training is crucial, especially considering growing demands and modern regulations. For example, Italian law no. 24/2017 sets forth a number of objectives regarding patient safety, risk management, and healthcare workers’ liability, with emphasis placed on continuous training [38].

It should be highlighted that there is often a lack of reliable data for developing models related to unaccompanied minors, which is challenging for pediatricians. Furthermore, migration patterns change over time, as observed in the case of Eastern European children during the Ukraine war, making predictions difficult. Data collection is also hindered by the dispersion of information among various institutions and registries, including prosecutors’ offices, local health units, and prefectures. Additionally, age assessment conducted by pediatricians or specialists can serve different purposes and data may be collected in different databases. The lack of uniformity in measurement protocols and the use of methods which are not always suitable for forensic contexts present additional challenges.

From a scientific viewpoint, radiation doses applied during these examinations are low enough to be ethically acceptable. Current non-radiographic methods for ascertaining whether or not a subject is 14 years old are imprecise since clinical and sexual evaluations cannot accurately determine age, partly due to possible developmental pathologies [39,40].

We believe that pediatricians should become familiar with the “age estimation” method proposed above, given the consequences that could result from incorrect age determination, both in clinical and criminological areas, especially in view of the implications of the 14-year-old threshold. A superficial approach and inaccurate estimation, beyond the consequences for the minors in question, could open profiles of liability for their pediatricians as well.

In Italy, as well as in other European countries, there is a need to establish regulatory frameworks that address certain ethical concerns linked to the use of ionizing radiation for age estimations in individuals who are minors or presumed to be so, particularly for forensic rather than clinical purposes [33,41]. This would entail providing hospital pediatricians and other specialists involved with a valid and reliable diagnostic tool, ultimately benefiting both the individual and society as a whole.

### Limits

It is important to recognize and discuss the limitations of this study, which may impact the interpretation of the results and the generalizability of the conclusions. Listed below are the main limitations of the study:

Limited data sources: The study relies on data from specific samples, one composed of Italian children aged between 5 and 15 years and the other made up of Chilean individuals aged between 11 and 22 years. These samples may not be representative of the general population, and the use of such samples may limit the generalizability of the results to other ethnic or geographic groups.

Orthodontic treatment: Study participants had been undergoing orthodontic treatment, which could have an impact on dental development. Orthodontic therapies can influence the timing and pattern of tooth development, potentially introducing bias into the age estimation method.

Radiation exposure: The proposed age estimation method involves exposing individuals to radiation, raising ethical concerns, especially when dealing with minors. Although the study mentions the importance of informed consent, it is crucial to carefully consider the potential risks associated with radiation exposure.

Disproportion between samples: The Chilean sample was much larger than the Italian one. In estimating age, variability may not be attributed exclusively to dental development; it may be influenced by factors related to ethnicity, genetics, or geographical origin.

In light of these aspects, future research should aim to address these limitations by incorporating more robust randomization procedures, ensuring larger and more balanced samples, and considering the influence of various demographic and environmental factors on dental development. Additionally, ethical considerations related to radiation exposure, especially in minors, should remain at the forefront of future investigations in this field.

## 5. Conclusions

In conclusion, our study has provided valuable insights into the age estimation of individuals, particularly focusing on the age of 14. We have established specific cut-off values based on dental mineralization, which have demonstrated high sensitivity and specificity for both males and females.

Over the past five decades, the need for precise and accurate age-determination methods has been a pressing concern. Traditional approaches, such as the assessment of skeletal development, have limitations, especially when the socio-economic and nutritional factors pertaining to the individual are unknown. Dental mineralization, on the other hand, offers a more reliable method since it is less influenced by external factors like diet and climate [30].

Our proposed method, which involves assessing the presence of open apices in specific teeth, has shown promise in accurately estimating whether or not an individual has attained 14 years of age. However, it is important to note that this method requires exposure to radiation, emphasizing the need for informed consent, especially when dealing with children and young individuals.

From a medico-legal perspective, age estimation in minors requires a collaborative effort among healthcare professionals, including pediatricians, neuropsychiatrists, and diagnostic imaging specialists. This multidisciplinary approach is crucial for accurate age determination and carries significant professional liability.

Challenges remain, particularly in countries like Italy, where the allocation of responsibility within the healthcare team is a topic of ongoing debate. Additionally, ethical concerns arise when radiological examinations are necessary for age estimation. While alternative methods like MRI exist, they are less practical and cost-effective [42].

It is important to underscore the importance of standardized strategies for age estimation, given the profound implications it holds within the realm of justice. Adequate training of personnel is essential to ensure the accurate implementation of age estimation techniques, aligning with modern regulations and safety standards.

Indeed, it is evident that many pediatricians may not be adequately prepared to perform such assessments on minors, whether for clinical or forensic purposes. This raises the question of insurance coverage in the event of serious errors, prompting the inquiry as to whether age estimation falls within the scope of regular pediatric practice or requires special consideration, especially given the individual responsibilities placed on every party involved in the assessment, which is tasked with providing highly specialized expertise. This is particularly relevant in countries like Italy, where the law distinguishes liabilities between professional figures [43]. It may also necessitate an increase in insurance premiums for forensic practice in this field, highlighting the challenges and legal implications associated with age estimation using dental radiographs.

## Figures and Tables

**Figure 1 healthcare-11-03047-f001:**
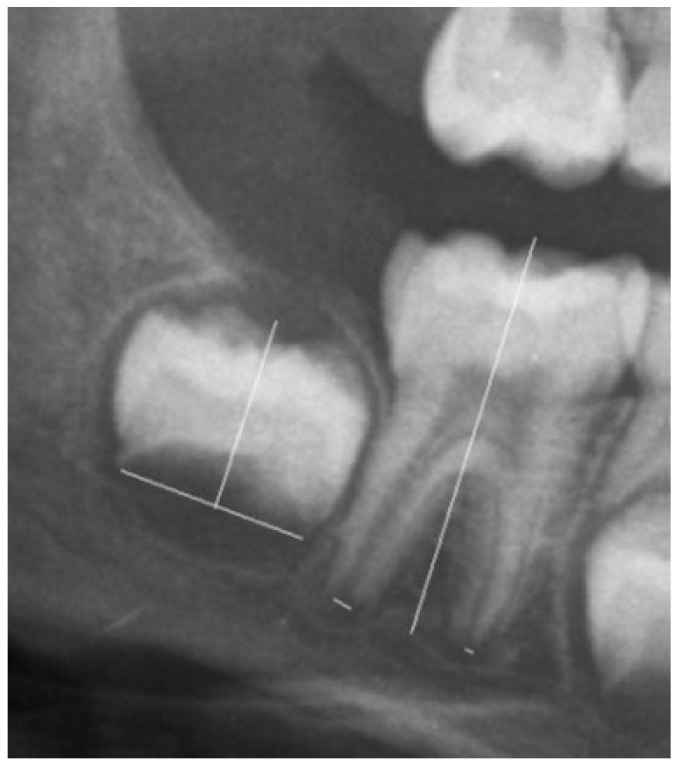
An example of tooth measurement with one or two roots.

**Table 1 healthcare-11-03047-t001:** Sensitivity and specificity for cut-off values for males and females at 14 years with two teeth with open apices (third molar excluded).

Sex	Males	Females
Cut-off	0.101	0.113
Sensitivity	80 (65–91)	83 (61–95)
Specificity	89 (82–94)	92 (86–96)
Correct Classification	87 (80–92)	91 (85–95)
Positive Predictive Value	88 (81–93)	91 (85–95)

**Table 2 healthcare-11-03047-t002:** Sensitivity and specificity for cut-off values for males and females at 14 years with just second and third molars not fully developed.

	Males		Females	
Cut-off	Third Molar	Second Molar	Third Molar	Second Molar
	0.73	0.16	0.77	0.10
Sensitivity	84 (79–89)		81 (76–85)	
Specificity	90 (82–96)		80 (72–86)	
Correct Classification	86 (81–89)		80 (76–84)	
Positive Predictive Value	96 (93–98)		93 (90–95)	

## Data Availability

Data are contained within the article.

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
