# Peer review of "Liability and Medico-Legal Implications in Estimating the Likelihood of Having Attained 14 Years of Age in Pediatric Clinical Practice: A Pilot Study"

_healthcare, 2023, doi:10.3390/healthcare11233047_

Round 1

Reviewer 1 Report (New Reviewer)

Comments and Suggestions for Authors

Dear Authors, I have made a few comments to the paper.

I urge you to revise some of the sentences, namely on the discussion, as I have detected some that are identical to already published articles. Text plagiarism may easily happen if research topics overlap, hence extra care is necessary.

Some sentences also need improving.

Also,  although ethical consent was given, it is not clear to what was it given to - i.e., did the participants' parents understood that data was going to be used in publications? conferences, etc, etc.  

For these reasons, I have recommended MAJOR revision. 

Comments on the Quality of English Language

English is more than acceptable. However, some sentences could be improved.

Author Response

We express our gratitude to Reviewer 1 for their insightful comments, which we trust have contributed to enhancing the quality of our work. Regarding textual similarities, the entire manuscript has been thoroughly revised to eliminate any overlaps, and it is our hope that these have been rectified. The language of the manuscript has undergone a comprehensive revision by a native English speaker, and clarifications have been incorporated from lines 102 to 104.

Reviewer 2 Report (New Reviewer)

Comments and Suggestions for Authors

This article addresses the issue of age estimation in minors, which is a much-debated topic these days. As the authors explain, it is important to distinguish between forensic age, which is predicted by various components of biological age, and chronological age of individuals, as these may differ due to various factors. As the author further states, dental age seems to be more appropriate because it is less influenced by socioeconomic factors compared to skeletal age. The present study provides a method that may prove useful in practice to estimate the age of 14 years, which is the most widely accepted age of full criminal responsibility worldwide. Although valuable research has been done, there are some points that should be carefully revised, and some information should be added. Please read my recommendations below:

-            In the key words – the word “responsibility” is misspelled.

-            The section on materials and methods is not very well organized and some important information is missing. Is the number of subjects studied 191/822 after mentioning the exclusion criteria? In the Materials and Methods section, a reference is needed in lines 112-113, and the 7th and 8th paragraphs in the MM section (lines 110-113; 115-119) appear to be identical, please revise carefully. Also, in the Material and Method section, it should be clearly stated which teeth were analyzed, that the method proposed by the authors in this article is based on the findings of Cameriere 2006, 2007. In the abstract, the authors state that this is a retrospective analysis; in Material and Method, this statement is missing. Retrospective should also be mentioned in the MM section. If it is a retrospective study, perhaps another expression of ethical statement should be added. I wonder how the authors could obtain so many informed consent forms if this is a retrospective study.

-            Given the nature of the study, I suggest that the term pilot or preliminary study should be added to the title and highlighted in the article.

-            The exclusion criteria are clearly stated in the Methods section, so there is no need to repeat them in the study limitations.

-            It is a method suggested by the authors, not a methodology. Please distinguish these two terms, method and methodology.

-            I realize that sex and ethnic differences in tooth development and eruption have already been studied, but in the context of the present study, these results should be mentioned at least briefly in the discussion section. So, did the present study compare sex differences in the proposed method for dental age? Were both ethnic variants analyzed together? Reference should be made to studies that have not demonstrated an effect of ethnic variety. I understand that you are referring to a study that mentions a smaller effect of climate on tooth age, but in the present study it is at least possible to compare these two groups from different countries to determine possible differences, if any.

-            I understand that the authors want to propose a unique and specific method for determining dental age that can be used in practice. But in this case, there should be more discussion about the “defense" of their proposed method. For example, the question “Why not use the previous methods?” should be answered at least partially, e.g., how this method differs from the one proposed by Camerie et al. 2006, 2007, why this method should be better and easier to use. Then, in the discussion, I propose to briefly address the widely used Demirjian method and, on the other hand, briefly address the method proposed by Gambier (which is based on eruption stages/phases and does not involve ionization), e.g. the method proposed by Gambier based on dental eruption (Gambier A, Rérolle C, Faisant M, Lemarchand J, Paré A, Saint-Martin P (2019) Contribution of third molar eruption to the estimation of the forensic age of living individuals.) and its use in practice with caution (Švábová Nee Uhrová P, Beňuš R, Chovancová Nee Kondeková M, Vojtušová A, Novotný M, Thurzo A. Use of third molar eruption based on Gambier's criteria in assessing dental age. Int J Legal Med.).

Comments on the Quality of English Language

Minor editing of English language required

Author Response

We express our gratitude to the Reviewer for their careful observations, which we hope have contributed to improving the text. The term 'responsibility' has been corrected and replaced with 'liability.' The organization of the Materials and Methods section has been modified. The duplicated sentence was an error in the text that we have addressed. The requested additions in the Materials and Methods and Discussion sections are highlighted in red in the text. The reference to an additional retrospective study has been removed. Repetitions regarding exclusion criteria have been eliminated. Wherever 'methodology' appeared, it has been corrected to 'method.' Explanations of the Demirjian and Gambier methods have been added. The works suggested by the reviewer have also been cited.

Reviewer 3 Report (New Reviewer)

Comments and Suggestions for Authors

I have read this paper with interest and I think the subject is original and very well presented and described. I also think that the method proposed by authors should be implemented because it could be very useful for different purposes, specially in medico-legal field.

Author Response

We thank the reviewer for their thoughtful considerations of our work.

Reviewer 4 Report (New Reviewer)

Comments and Suggestions for Authors

Dear Editor of Healthcare and Authors,

the article is interesting and well-written, and believe it to have quality to warrant publication in this journal. There is only one small amendment: the authors state that "The dental age estimation method's robustness against external influences, like socio-economic factors, is noteworthy. It underscores the limitations of skeletal age methods, advocating for dental mineralization-based approaches." but nothing about this is researched in this paper. There aren't socioeconomic variables being related with the age estimation, and no comparison with skeletal methods. Thus, this statements need to be eliminated OR be contextualized from other literature sources.

Best regards.

Comments on the Quality of English Language

Moderate ammendments needed. 

Author Response

We thank the reviewer for their observation. The sentence has been corrected and amended. The changes have been highlighted in red.

Round 2

Reviewer 1 Report (New Reviewer)

Comments and Suggestions for Authors

Thank you for taking the time to address the issues raised.

Comments on the Quality of English Language

Quality was much improved.

Reviewer 2 Report (New Reviewer)

Comments and Suggestions for Authors The authors have taken all recommendations into account and revised the manuscript accordingly, so I am proposing the paper for publication.

Comments on the Quality of English Language

 Minor editing of English language required

This manuscript is a resubmission of an earlier submission. The following is a list of the peer review reports and author responses from that submission.

Round 1

Reviewer 1 Report

Comments and Suggestions for Authors

The article is interesting and well-written, but the research in question, although described as retrospective and observational, involves health data related to minors, which is why an ethics committee's opinion is necessary. The technical-scientific committee is not an ethics committee. Furthermore, it is unclear where the data used in the study are derived.

Reviewer 2 Report

Comments and Suggestions for Authors

The issue raised by the authors is highly topical and of great importance due to the important and growing migratory movements of recent times. On the other hand, it is well known that minors are not deprived of rights, although there are limits that delimit age ranges from which they should be considered. 

Despite this, the study has a series of important limitations. The main one is the methodological approach. In order to conclude the usefulness of this type of method, it is necessary that the observations have been made by at least two observers. This would make it possible to evaluate the intra- and inter-observer error and to assess whether it is really accurate and precise (e.g. ICC). It would also be necessary to specify how the measurements taken and the actual age of the subjects correlate. 

On the other hand, the results obtained are not specifically discussed with the existing literature. It is necessary to consider the variability in dental mineralisation and to conclude from the results obtained that this type of estimation requires the use of more than one method to give reliable results.